# DIFFUSION-ADVECTION TRANSFORMER FOR AIR QUALITY PREDICTION

## ABSTRACT

Air pollution is a major concern for public health and the environment globally, which highlights the need for effective monitoring and predictive modeling to mitigate its impact. Although data-driven models have shown promising results in air quality prediction, they still struggle to model the underlying physical mechanisms of pollutant dispersion, where diffusion governs small-scale spreading and advection drives large-scale directional transport. To address this limitation, we propose the Diffusion-Advection Transformer (DA-Transformer), a novel physics-informed architecture. Specifically, the model integrates the two key physical mechanisms by embedding diffusion and advection as differential equation-based components. These physics-informed modules are incorporated into a Transformer framework to enable the model to better capture pollutant transport dynamics, such as local diffusion-driven smoothing and wind-induced directional propagation in air quality data. Experiments on three real-world datasets demonstrate that DA-Transformer consistently outperforms baseline models in $PM_{2.5}$ concentration prediction and achieves substantial gains over its variants that exclude diffusion and advection in their model design.

## 1 INTRODUCTION

Air pollution is a critical global environmental challenge with profound health and ecological implications. According to the World Health Organization (WHO) in their fact sheet on ambient (outdoor) air quality and health, the combined effects of ambient air pollution and household air pollution are associated with 6.7 million premature deaths annually(WHO, 2024). The situation is especially acute in urban regions, particularly in many megacities where annual average fine particulate matter ($PM_{2.5}$) concentrations exceed the WHO 2021 guideline of 5 μg/m³ by approximately 4–9×; recent assessments report citywide means around 30–40 μg/m³ (Energy Policy Institute at the University of Chicago, 2024; Centre for Research on Energy and Clean Air, 2023). High concentrations of air pollutants such as sulfur dioxide ($SO_2$), nitrogen dioxide ($NO_2$), and ozone ($O_3$) are associated with respiratory diseases, cardiovascular conditions, and lung cancer in humans(Arbex et al., 2012). Moreover, these pollutants adversely impact the environment, contributing to phenomena like acid rain and agricultural degradation(Bhargava & Bhargava, 2013).

Air quality prediction is inherently challenging due to the complex spatiotemporal nature of air pollution, where pollutant levels are influenced by an interplay of multiple factors such as meteorological conditions (e.g., temperature, wind speed, wind direction), industrial emissions, and traffic patterns(Bai et al., 2018). These factors interact in complex ways across both space and time, posing significant challenges for accurate air quality modeling(Zhang et al., 2012). Traditional statistical approaches, such as Historical Average (HA)(Smith & Demetsky, 1997) and Vector Autoregression (VAR)(Sims, 1980), often struggle to fully capture these intricate relationships. HA fails to account for dynamic changes in pollution driven by weather and human activities, while VAR is limited in modeling nonlinear interactions and becomes computationally inefficient when extended to high-dimensional spatiotemporal data.

In recent years, machine learning (ML) techniques have been increasingly applied to air quality prediction(Iskandaryan et al., 2020). Neural networks, particularly recurrent neural networks (RNNs)(Rumelhart et al., 1986) and their variants such as long short-term memory networks (LSTMs)(Hochreiter, 1997), have shown promise in analyzing time-series air quality data. How-

ever, RNNs suffer from the vanishing gradient problem, which makes it difficult to learn long-term dependencies in the data, especially when dealing with extended air quality time-series data(Kim et al., 2018). LSTMs mitigate this issue to some extent but still struggle with very long-range dependencies and require careful tuning of hyperparameters(Kandadi & Shankarlingam, 2025). Convolutional neural networks (CNNs)(LeCun et al., 1998) have been employed to extract spatial features from air quality maps. Although CNNs are effective at capturing local spatial features, they struggle to model long-range dependencies(Liu et al., 2023), for example, assessing the influence of emissions advected from a distant area. Graph neural networks (GNNs)(Sperduti & Starita, 1997) and their specific type, graph convolutional networks (GCNs)(Kipf & Welling, 2016), have emerged as powerful tools in air quality prediction. GNNs are designed to handle data with graph-structured relationships, which are highly relevant in air quality modeling(Han et al., 2022). However, constructing an accurate graph structure for air quality data is challenging(Ferrer-Cid et al., 2021). The relationships between different geographical locations in terms of pollutant dispersion are complex and may not be accurately represented by a simple graph(Zhu et al., 2020). Moreover, GNNs are computationally expensive, especially when dealing with large-scale graphs representing extensive geographical areas(Wu et al., 2020). The Transformer(Vaswani, 2017) architecture has also been introduced to air quality prediction, to capture relationships among different temporal and spatial features, such as pollutant concentrations across locations and time(Zhang & Zhang, 2023; Yu et al., 2023). However, these data-driven approaches inadequately account for the underlying physical processes governing air pollution, thereby limiting their predictive performance(Liu et al., 2022).

To overcome these challenges, recent studies have explored hybrid models that combine physical principles with machine learning techniques. Specifically for air-quality modeling, diffusion describes local dispersion driven by turbulent mixing, whereas advection represents wind-driven directional transport of pollutants. Recent work such as AirPhyNet(Hettige et al., 2024) and Air Dual-ODE(Tian et al., 2024), integrate diffusion and advection with neural architectures through, respectively, graph-based operators and dual-branch neural ordinary differential equations. Building on these ideas, we embed diffusion and advection as discrete, learnable operators directly within a Transformer, avoiding explicit graph construction and continuous-time solvers while keeping the formulation compatible with discrete monitoring measurements.

To summarize, this paper makes the following contributions:

1. We design a physics-informed module implementing diffusion and advection within a Transformer-based spatiotemporal forecasting framework for air quality prediction. Diffusion is computed by applying a temperature-dependent coefficient to the Laplacian of input features, while advection is modeled as the product of wind velocity components and spatial gradients. These physics-informed modules are incorporated into the training process and jointly optimized with model parameters.

2. We introduce learnable discretization that bridges continuous diffusion–advection partial differential equations (PDEs) and discrete station–time observations. The diffusion term adopts a discrete Laplacian weighted by a learnable, temperature-dependent coefficient, and the advection term uses directional differences modulated by wind velocity fields. This design enables end-to-end training inside a Transformer and avoids external PDE solvers or pre-defined graphs. It uses a numerically stable local neighborhood update and the computation scales linearly with the number of spatial sites.

3. We evaluate the proposed model across three representative regions that span distinct emission and meteorological regimes. Beijing is a dense megacity with severe $PM_{2.5}$ levels during our study window, driven by seasonal heating and frequent winter inversions. The UK represents a maritime climate where synoptic advection dominates. California combines coastal and inland basins with complex topography and episodic wildfires. The California dataset is newly constructed by us from raw pollutant and meteorological records. Across 24/48/72-hour horizons, the proposed DA-Transformer consistently improves MAE and RMSE over representative baselines, including statistical (HA, VAR), sequence models (RNN, LSTM), Transformer variants (Transformer, Informer, EMD-Transformer-BiLSTM), and graph-based spatiotemporal models (DCRNN, GTS, AirPhyNet).

## 2 PRELIMINARIES

### 2.1 PROBLEM STATEMENT

The goal of air quality prediction is to estimate future pollutant levels using historical observations from $N$ monitoring stations. The readings from $N$ air quality monitoring stations at a given time $t$ are represented as $X_t \in \mathbb{R}^{N \times D}$, where $D$ is the number of attibutes, such as air pollutant concentrations (e.g., PM$_{2.5}$, NO$_2$) and meteorological data (e.g., temperature, wind speed). Here, $x_{ij}$ represents the $j$-th feature observed at station $i$. The goal is to design $f(\cdot)$ that predicts air quality attibutes for the next $\tau$ time steps based on the historical data from the past $T$ time steps. This can be formalized as:

$$Y_{T:(T+\tau)} = f(X_{1:T}), \tag{1}$$

where $X_{1:T} \in \mathbb{R}^{N \times T \times D}$ represents the historical data, and $Y_{T:(T+\tau)} \in \mathbb{R}^{N \times \tau \times D}$ denotes the predicted future data.

### 2.2 RELATED WORK

Air quality prediction methods are commonly grouped into process-based numerical models and data-driven methods. Process-based models describe pollutant transport by solving differential equations grounded in atmospheric dynamics (Vardoulakis et al., 2003; Daly & Zannetti, 2007), whereas data-driven models learn complex spatiotemporal structure directly from historical observations (Yi et al., 2018; Zhou et al., 2021).

Recent research has focused on integrating physical priors into deep learning models to enhance generalization and interpretability. Several studies incorporate physics either through regularized loss functions (Jia et al., 2019), constrained optimization (Peng & Karpatne, 2022), or neural ODE formulations (Tian et al., 2024). AirPhyNet (Hettige et al., 2024), for instance, embeds advection-diffusion dynamics into GNNs, while other works rely on graph structures or predefined PDE solvers.

Distinct from prior work, our DA-Transformer integrates temperature-dependent diffusion and wind-driven advection as learnable, differentiable operators within the Transformer and optimizes them jointly. This preserves the representational power of attention while modeling physical transport end-to-end. To our knowledge, this is the first air quality forecasting models to embed diffusion and advection directly inside a Transformer without explicit graph construction or external PDE solvers.

### 2.3 DIFFUSION-ADVECTION EQUATION

The diffusion–advection equation describes the interplay of diffusion and advection in continuum transport. A canonical one-dimensional form (derivable from mass conservation together with Fick's law)(Fourier, 1822; Fick, 1855; Bird et al., 2002; Patankar, 1980) is:

$$\frac{\partial u}{\partial t} + v(x,t)\frac{\partial u}{\partial x} = D\frac{\partial^2 u}{\partial x^2}, \tag{2}$$

where:

- $u(x,t)$: scalar quantity (e.g., concentration, temperature).
- $v(x,t)$: advection velocity.
- $D$: diffusion coefficient.

The left-hand side captures temporal changes and advection effects, while the right-hand side accounts for diffusion processes. In two dimensions, a commonly used anisotropic form is:

$$\frac{\partial u}{\partial t} + v_x\frac{\partial u}{\partial x} + v_y\frac{\partial u}{\partial y} = D_x\frac{\partial^2 u}{\partial x^2} + D_y\frac{\partial^2 u}{\partial y^2}. \tag{3}$$

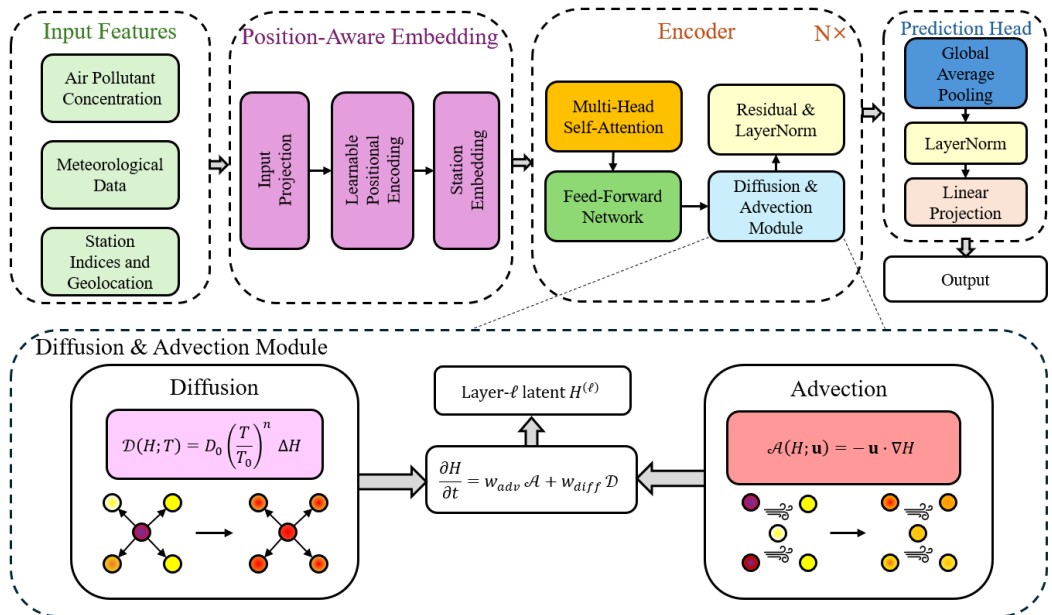

Figure 1: **DA-Transformer overview.** Top: pipeline from input features and position-aware embedding through an $N\times$ encoder stack to the prediction head. Bottom: expanded view of the Physics block inserted in each encoder layer, showing the diffusion and advection branches and their gated combination.

Standard atmospheric treatments present the corresponding conservation-form extensions and applications to reactive species; see, for example, Seinfeld and Pandis(Seinfeld & Pandis, 2016) and classic fluid-dynamics expositions(Batchelor, 1967).

## 3 METHODOLOGY

### 3.1 MODEL OVERVIEW

This section presents the proposed DA-Transformer, a physics-informed Transformer that couples self-attention with advection and diffusion operators for spatiotemporal air quality forecasting.

**Task and Representation.** Let $x \in \mathbb{R}^{S \times N \times C}$ denote a window of $S$ time steps over $N$ stations with $C$ variables (e.g., $PM_{2.5}$, $PM_{10}$, $O_3$, temperature, wind, latitude and longitude). The objective is to predict $\hat{y} \in \mathbb{R}^{K \times N}$ for horizons $1{:}K$.

**Encoding and Embeddings.** Each feature is projected to a shared latent dimension $d_{\text{model}}$ via a linear layer. We use learnable positional encodings to capture temporal ordering and non-periodic patterns.

**Physics-Guided Encoder.** The encoder is a stack of DA-Transformer layers. Each layer applies multi-head self-attention to couple long-range temporal signals across stations, followed by a position-wise feed-forward network with residual connections and layer normalization. To inject physical priors, every layer contains a differentiable physics module: diffusion is implemented as a discrete Laplacian on latent feature maps with a temperature-conditioned coefficient, and advection as wind-projected directional gradients along spatial axes. These operators are trained jointly with the network, encouraging representations that respect transport phenomena.

**Prediction Head (Non-Autoregressive).** Instead of an autoregressive decoder that iteratively rolls out future steps, we adopt a lightweight head that maps the encoded sequence to all $K$ horizons in one pass. Concretely, we apply global average pooling over the temporal dimension at each

station to obtain $N$ station-wise context vectors, then use a normalization layer and a station-wise linear projection to produce $\hat{y} \in \mathbb{R}^{K \times N}$. This head is simplified because it removes the decoder stack, cross-attention, teacher forcing, and iterative rollout; the computational cost is dominated by pooling $O(SNd_{\text{model}})$ and a single projection $O(Nd_{\text{model}}K)$ rather than $O(K)$ decoder passes with attention layers. Forecasting accuracy is maintained because the encoder already aggregates cross-time and cross-station context via self-attention, the physics module supplies inductive bias that captures diffusion-driven smoothing and wind-driven transport, and the one-shot mapping avoids horizon-wise error accumulation.

**Summary.** The architecture links data-driven sequence modeling with physically grounded transport, yielding interpretable and competitive forecasts with a simple and efficient prediction head.

## 3.2 LEARNABLE POSITIONAL ENCODING

In Transformer models, positional encoding supplies the ordering information that the architecture itself does not encode. The model has no built-in notion of position because it contains neither recurrence nor convolution. Positional encoding therefore provides the positional context needed to distinguish locations across the input sequence.

Unlike the original Transformer design, which employs fixed sinusoidal functions, our model adopts a learnable positional encoding strategy. Each time step is assigned a trainable embedding vector, forming a tensor $\text{PE} \in \mathbb{R}^{S \times 1 \times 1 \times d_{\text{model}}}$ that is optimized jointly with other model parameters. This design enables the model to flexibly capture temporal dynamics based on real data, rather than relying on predefined frequency bases.

Learnable positional encoding is particularly advantageous for air quality forecasting, where temporal patterns are influenced by both periodic cycles (e.g., diurnal and seasonal trends) and instant meteorological changes. By adapting the temporal representation to the data distribution, the model can better identify long-term dependencies and temporally salient events, improving its ability to forecast pollutant evolution under complex and dynamic conditions.

## 3.3 SPATIAL AND TEMPORAL ATTENTION

### 3.3.1 SPATIAL ATTENTION

Spatial attention aggregates information from all monitoring stations within the input window by applying self-attention over flattened spatiotemporal tokens. Each sample provides geocoordinates (latitude and longitude) and meteorological fields (e.g., wind and temperature), allowing the model to learn geometry- and flow-aware patterns without an explicit graph.

Let $F \in \mathbb{R}^{S \times N \times C}$ denote features over $S$ time steps and $N$ stations (batch dimension suppressed). We flatten the temporal and spatial axes to obtain a sequence $Z \in \mathbb{R}^{(S \cdot N) \times C}$, apply multi-head self-attention to $Z$, and then reshape the result back to $(S, N, C)$ for subsequent layers. Concretely, with $Q = ZW_Q$, $K = ZW_K$, and $V = ZW_V$, the spatial-attention output is

$$\text{MHA}(Z) = \text{softmax}\left(\frac{QK^\top}{\sqrt{d_k}}\right) V,$$

which mixes information across stations and times within the window before being reshaped to $(S, N, C)$.

### 3.3.2 TEMPORAL ATTENTION

Temporal attention serves as an additional self-attention block over the same flattened spatiotemporal sequence. Although it uses the same operator as above, its separate parameters and the presence of positional encodings encourage sensitivity to temporal regimes such as rush-hour patterns, temperature inversions, or weather transitions.

Formally, we reuse the flattened sequence $Z \in \mathbb{R}^{(S \cdot N) \times C}$ and apply another multi-head self-attention layer with independent projections $W_Q^{(\tau)}, W_K^{(\tau)}, W_V^{(\tau)}$:

$$\mathrm{MHA}_\tau(Z) = \mathrm{softmax}\left( \frac{Z W_Q^{(\tau)} (Z W_K^{(\tau)})^\top}{\sqrt{d_k}} \right) (Z W_V^{(\tau)}).$$

The output is reshaped back to $(S, N, C)$ and passed forward. In practice, this global mixing lets the model emphasize those spatiotemporal tokens that are most informative for forecasting under changing meteorological conditions.

### 3.4 DIFFUSION AND ADVECTION

#### 3.4.1 DIFFUSION

We implement a temperature-conditioned diffusion block inside each DA-Transformer layer, realized as a five-point discrete Laplacian whose strength depends on ambient temperature. Concretely, the diffusion coefficient is set as a temperature function

$$D(T) = D_0 \left( \frac{T}{T_0} \right)^n, \tag{4}$$

where $D_0$ and $n$ are fixed hyperparameters and $T_0$ is a reference temperature (Kelvin), matching our implementation.

Let $H^{(\ell)} \in \mathbb{R}^{B \times S \times N \times d}$ be the latent entering layer $\ell$ (batch $B$, window length $S$, stations $N$, channels $d$). Indices are $b \in \{0, \ldots, B-1\}$, $s \in \{0, \ldots, S-1\}$, $i \in \{0, \ldots, N-1\}$, and $c \in \{0, \ldots, d-1\}$. We adopt periodic boundary conditions along both the station axis $i$ and the channel axis $c$ (i.e., $i \pm 1$ is taken modulo $N$ and $c \pm 1$ modulo $d$). Assuming unit grid spacing, the five-point discrete Laplacian is:

$$\left[ \Delta H^{(\ell)} \right]_{b,s,i,c} = H_{b,s,i+1,c}^{(\ell)} + H_{b,s,i-1,c}^{(\ell)} + H_{b,s,i,c+1}^{(\ell)} + H_{b,s,i,c-1}^{(\ell)} - 4 H_{b,s,i,c}^{(\ell)}. \tag{5}$$

The diffusion contribution is then

$$\mathcal{D}_{b,s,i,c} = D\big(T_{b,s}\big) \left[ \Delta H^{(\ell)} \right]_{b,s,i,c}, \tag{6}$$

which is injected via a residual update inside the layer.

#### 3.4.2 ADVECTION

To capture advective transport, a differentiable operator is embedded in each DA-Transformer layer, transporting latent features along the observed wind field. The two-dimensional wind is represented by eastward ($u$) and northward ($v$) components from reanalysis or station measurements. When only wind speed $W$ and wind direction $\theta$ are available, we use the meteorological from-direction convention (clockwise from true north, indicating the origin of the wind) to obtain the components

$$u = -W \sin(\theta), \qquad v = -W \cos(\theta),$$

and standardize units (speeds in m·s$^{-1}$; directions converted to radians for trigonometric functions). Because monitoring stations are irregularly located in latitude–longitude space, we order stations by geographic proximity and apply differences along that order so that the station-axis derivative acts as a local spatial-derivative surrogate.

Let $H \in \mathbb{R}^{B \times S \times N \times d}$ be the latent feature field (batch $B$, window length $S$, stations $N$, channels $d$). We approximate directional derivatives with first-order forward differences and a zero-gradient condition at the first index:

$$(\delta_{\mathrm{stat}} H)_{b,s,i,c} = H_{b,s,i,c} - H_{b,s,i-1,c}, \qquad (\delta_{\mathrm{stat}} H)_{b,s,0,c} = 0, \tag{7}$$

$$(\delta_{\mathrm{feat}} H)_{b,s,i,c} = H_{b,s,i,c} - H_{b,s,i,c-1}, \qquad (\delta_{\mathrm{feat}} H)_{b,s,i,0} = 0. \tag{8}$$

The advection contribution then follows the standard transport form $-\mathbf{v} \cdot \nabla$:

$$\mathcal{A}_{b,s,i,c} = -\Big( u_{b,i,c} \, (\delta_{\mathrm{feat}} H)_{b,s,i,c} + v_{b,i,c} \, (\delta_{\mathrm{stat}} H)_{b,s,i,c} \Big), \tag{9}$$

which is added through the residual stream in each layer. Equation equation 9 uses first-order backward (one-sided) finite differences with a zero-gradient condition at the leading index on each axis, which is a simple and stable scheme for convective transport (Patankar, 1980; Bird et al., 2002).

### 3.4.3 LEARNABLE INTEGRATION OF ADVECTION AND DIFFUSION

While both advection and diffusion reflect distinct physical processes, their relative influence on pollutant transport may vary across different meteorological or geographical contexts. To adaptively model these variations, we introduce two learnable scalar weights: $w_a$ and $w_d$, which represent the relative importance of advection and diffusion, respectively. These parameters are trained jointly with the rest of the model to enable dynamic adjustment based on the data distribution.

To ensure stable training, $w_a$ and $w_d$ are initialized using Xavier initialization (Glorot & Bengio, 2010), which mitigates vanishing or exploding gradients and accelerates convergence in deep networks.

Let $\mathcal{A}_{\mathbf{u}}(\cdot)$ and $\mathcal{D}_T(\cdot)$ denote the advection and diffusion operators, respectively; both operators map $\mathbb{R}^{B \times S \times N \times d} \to \mathbb{R}^{B \times S \times N \times d}$. The weighted integration of the advection and diffusion terms is expressed as:

$$H_{\text{out}}^{(\ell)} = H^{(\ell)} + w_a \, \mathcal{A}_{\mathbf{u}}\big(H^{(\ell)}\big) + w_d \, \mathcal{D}_T\big(H^{(\ell)}\big). \tag{10}$$

This mechanism allows the model to automatically balance the contribution of each physical process. For instance, under stable atmospheric conditions with calm wind, the model may learn to assign a higher weight to the diffusion term, emphasizing local dispersion. During strong wind events, the advection term would receive a greater weight to reflect the dominant role of wind-driven pollutant transport.

By enabling this learnable integration, the model becomes more flexible and context-aware, improving its ability to simulate spatiotemporal variations in air quality across diverse environmental conditions. This design further strengthens the physical interpretability of the model while enhancing predictive performance.

## 4 EXPERIMENT

### 4.1 DESCRIPTION OF THE DATA

To evaluate the effectiveness and generalizability of our model, we adopt three hourly air-quality datasets spanning distinct regions and regimes. Complete station lists and variable definitions are provided in Appendix A.3 (Tables 3–10).

**Beijing, China.** (2013-03-01 to 2017-02-28): 12 nationally controlled stations ($PM_{2.5}$, $PM_{10}$, $SO_2$, $NO_2$, CO, $O_3$) and meteorology (temperature, pressure, dew-point, precipitation, wind direction/speed). Source: UCI "Beijing Multi-site Air-Quality Data".[1]

**United Kingdom.** (2015-01-01 to 2023-11-07): 159 stations from the national AURN network with $PM_{2.5}$, $PM_{10}$, $NO_x$, $O_3$ and meteorology (temperature, wind direction/speed).[2] The UK features a *temperate oceanic (Köppen Cfb)* climate with strong maritime influence and both coastal and inland urban areas, offering a complementary contrast to Beijing.

**California, United States.** (2021-01-01 to 2023-12-31): a new dataset created and curated by us, comprising 62 stations constructed by merging OpenAQ air-pollution data with ERA5 reanalysis meteorology (temperature, wind speed/direction, station coordinates).[3][4] The region combines complex topography (coast, Central Valley, Sierra Nevada), strong diurnal wind regimes (e.g., sea-breeze, mountain–valley flows), and episodic wildfire smoke events (e.g., the 2021 Dixie and Caldor fires; the 2022 Mosquito fire), making it a novel testbed for robustness to extreme, rapidly evolving pollution episodes.

---

[1] http://archive.ics.uci.edu/dataset/501/beijing+multi+site+air+quality+data

[2] https://www.kaggle.com/datasets/airqualityanthony/uk-defra-aurn-air-quality-data-2015-2023

[3] OpenAQ: https://openaq.org/ (API: https://api.openaq.org/)

[4] ERA5: https://cds.climate.copernicus.eu/datasets/reanalysis-era5-single-levels

Following common practice in the literature, we take fine particulate matter ($PM_{2.5}$) as the primary prediction target, while using other pollutant channels and meteorology as covariates.

## 4.2 EXPERIMENT SETTINGS

### 4.2.1 BASELINES

We evaluate ten baselines spanning four families: (i) classical statistical models (HA, VAR(Toda, 1991)); (ii) neural sequence models (RNN(Rumelhart et al., 1986), LSTM(Hochreiter, 1997)); (iii) Transformer-based models (Transformer(Vaswani, 2017), Informer(Zhou et al., 2021), EMD-Transformer-BiLSTM(Dong et al., 2024)); and (iv) graph-based spatiotemporal approaches (DCRNN(Li et al., 2018), GTS(Shang et al., 2021), AirPhyNet(Hettige et al., 2024)).

### 4.2.2 IMPLEMENTATION DETAILS

We implement our model using PyTorch 2.5.1 on an NVIDIA GeForce RTX 3080 GPU. The AdamW optimizer is used with a learning rate of 1e-4 and a weight decay of 1e-4. A warm-up strategy is applied, followed by a cosine annealing schedule with warm restarts ($T_0 = 5$, $T_{mult} = 2$). Mixed precision training is enabled using automatic mixed precision (AMP).

Our DA-Transformer consists of 3 encoder layers and 2 decoder layers, with a hidden dimension $d_{model} = 64$ and 8 attention heads. A 24-hour historical input window is used to predict PM2.5 concentrations over horizons of 24, 48, and 72 hours. Input features, including meteorological and spatial variables, as well as the target variable, are normalized using `StandardScaler`.

### 4.2.3 EVALUATION METRICS

We report two standard error metrics including Mean Absolute Error, MAE; Root Mean Square Error, RMSE computed over all test samples (across stations and horizons unless stated otherwise): $\text{MAE} = \frac{1}{n} \sum_{i=1}^{n} |y_i - \hat{y}_i|$ and $\text{RMSE} = \sqrt{\frac{1}{n} \sum_{i=1}^{n} (y_i - \hat{y}_i)^2}$. MAE reflects the average absolute deviation, whereas RMSE penalizes larger errors more strongly. In both metrics, lower values indicate better performance.

## 4.3 PERFORMANCE COMPARISON

Table 1 presents the performance of our model compared to all baselines on three datasets. Accordingly, DA-Transformer outperforms all competing baselines in both metrics on Beijing, UK and California datasets. The results highlight the effectiveness of integrating physical principles with Transformer-based architectures to capture complex relationships.

Table 1: Performance comparison of models with 24-hour input for different prediction horizons. Lower is better.

| Region | Beijing | | | | | | UK | | | | | | California | | | | | |
|---|---|---|---|---|---|---|---|---|---|---|---|---|---|---|---|---|---|---|
| Horizon | 24h | | 48h | | 72h | | 24h | | 48h | | 72h | | 24h | | 48h | | 72h | |
| Metric | MAE | RMSE | MAE | RMSE | MAE | RMSE | MAE | RMSE | MAE | RMSE | MAE | RMSE | MAE | RMSE | MAE | RMSE | MAE | RMSE |
| HA | 64.50 | 81.52 | 64.64 | 81.60 | 64.72 | 81.69 | 4.04 | 6.33 | 4.04 | 6.33 | 4.04 | 6.33 | 17.55 | 21.66 | 17.55 | 21.66 | 17.55 | 21.66 |
| VAR | 64.05 | 80.21 | 66.91 | 81.56 | 67.37 | 82.34 | 4.07 | 6.38 | 2.87 | 4.99 | 2.89 | 5.01 | 17.63 | 21.75 | 16.60 | 20.64 | 16.74 | 20.79 |
| RNN | 53.42 | 72.86 | 57.43 | 76.81 | 57.86 | 78.15 | 5.11 | 7.94 | 6.97 | 8.03 | 7.15 | 8.20 | 19.61 | 27.53 | 20.86 | 28.46 | 21.05 | 28.62 |
| LSTM | 51.76 | 71.02 | 54.62 | 72.85 | 57.01 | 75.25 | 2.89 | 4.85 | 3.87 | 5.60 | 3.96 | 5.75 | 19.69 | 28.43 | 19.86 | 28.70 | 20.18 | 28.83 |
| Transformer | 51.78 | 71.11 | 54.71 | 73.16 | 56.12 | 74.23 | 3.09 | 5.01 | 3.63 | 5.50 | 3.89 | 5.82 | 19.14 | 28.57 | 20.47 | 29.31 | 20.76 | 30.01 |
| Informer | 30.41 | 44.81 | 37.19 | 51.22 | 45.36 | 59.72 | 3.48 | 5.66 | 7.09 | 10.58 | 11.32 | 15.47 | 13.59 | 19.21 | 21.99 | 26.78 | 34.13 | 40.16 |
| EMD-Tr-BiLSTM | 74.70 | 95.07 | 75.37 | 97.43 | 75.42 | 98.23 | 4.03 | 6.13 | 4.67 | 6.95 | 4.81 | 7.15 | 15.75 | 23.16 | 17.41 | 25.01 | 19.71 | 27.71 |
| DCRNN | 61.16 | 81.80 | 73.47 | 97.01 | 73.79 | 97.99 | 3.42 | 5.75 | 6.11 | 9.74 | 9.33 | 13.59 | 13.97 | 19.22 | 14.66 | 20.20 | 16.44 | 22.33 |
| GTS | 64.21 | 81.84 | 69.72 | 93.33 | 70.41 | 93.96 | 3.31 | 5.66 | 5.88 | 9.36 | 8.96 | 12.97 | 13.73 | 19.86 | 14.55 | 20.42 | 16.28 | 22.56 |
| AirPhyNet | 30.94 | 40.53 | 62.80 | 70.73 | 61.96 | 70.58 | 3.29 | 4.95 | 3.58 | 4.85 | 3.78 | 5.17 | 11.77 | 15.89 | **13.81** | 17.89 | **14.68** | 18.93 |
| Our Model | **25.04** | **30.16** | **28.81** | **33.91** | **29.45** | **34.85** | **2.76** | **3.69** | **2.73** | **3.70** | **2.75** | **3.71** | **11.59** | **15.12** | 13.88 | **17.21** | 14.75 | **18.01** |

From Table 1, we can draw several observations:

1) Deep learning-based approaches significantly outperform classical baselines such as HA and VAR, demonstrating their capability in modeling complex and nonlinear temporal dependencies for air quality forecasting.

Table 2: Ablations on physics transport and positional encodings. Lower is better. $\Delta$ denotes the increase vs. the full model.

| Dataset | Full model | | w/o advection–diffusion | | w/o positional encodings | |
|---|---|---|---|---|---|---|
| | MAE | RMSE | MAE ($\Delta$) | RMSE ($\Delta$) | MAE ($\Delta$) | RMSE ($\Delta$) |
| Beijing | 29.45 | 34.85 | 30.25 (+0.80, 2.7%) | 35.16 (+0.31, 0.9%) | 30.10 (+0.65, 2.2%) | 35.08 (+0.23, 0.7%) |
| UK | 2.75 | 3.71 | 2.76 (+0.01, 0.4%) | 3.71 (+0.00, 0.0%) | 2.76 (+0.01, 0.4%) | 3.71 (+0.00, 0.0%) |
| California | 14.75 | 18.01 | 14.81 (+0.06, 0.4%) | 18.06 (+0.05, 0.3%) | 16.00 (+1.25, 8.5%) | 19.15 (+1.14, 6.3%) |

2) Transformer-based models, including the vanilla Transformer, Informer, and EMD-Transformer-BiLSTM, generally achieve better performance than RNN and LSTM. This highlights the effectiveness of attention mechanisms in capturing long-range temporal relationships. However, these methods do not explicitly model spatial structures or incorporate physical priors, which can limit their performance under complex environmental conditions.

3) Graph-based models such as DCRNN, GTS, and AirPhyNet perform competitively by modeling inter-station spatial dependencies. In particular, AirPhyNet benefits from integrating advection-diffusion mechanisms within a graph neural network, achieving robust results especially on the Beijing dataset.

4) Our proposed DA-Transformer achieves the best overall performance across most datasets and forecasting horizons. While it does not dominate every individual case, it consistently provides strong and stable results. Across 24/48/72-h horizons, gains are uniform on UK, most pronounced at 72 h on Beijing, and concentrated at 24 h on California (with RMSE advantages persisting at 48–72 h), indicating that physics-guided modules improve stability while the largest gains vary by region.

Overall, the results demonstrate that DA-Transformer successfully combines data-driven learning with physically grounded modeling, enabling robust air quality forecasting across varied environmental scenarios.

## 4.4 ABLATION STUDY

**Effect of Physics knowledge.** We remove the advection–diffusion transport operators and keep all other components unchanged. As shown in Table 2, the largest and most consistent gains from explicit transport appear in *Beijing* (MAE $-0.80$ / 2.7%, RMSE $-0.31$ / 0.9%), where directional flow and local mixing are more pronounced; *California* and *UK* exhibit only marginal differences, suggesting weaker cross-station transport in our window or that attention already captures most residual coupling when flow signals are mild.

**Effect of Positional Encodings.** The second test removes learnable positional encodings while keeping the rest intact (Table 2). *California* is strongly sensitive (MAE $+1.25$ / 8.5%, RMSE $+1.14$ / 6.3%), consistent with pronounced diurnal/weekly cycles and coastline-driven regime shifts; *Beijing* benefits as well but less than from dual dynamics, indicating complementary roles—positional encodings stabilize temporal phase, while advection–diffusion injects physically grounded spatial coupling. *UK* changes are negligible, implying saturation under our setup.

## 5 CONCLUSION AND FUTURE WORK

This paper proposes DA-Transformer, a physics-informed neural network that leverages fundamental physical equations and seamlessly integrates them into a deep learning architecture. By combining transformer with diffusion and advection, our model achieves a seamless integration of physical modeling and data-driven learning, demonstrating excellent performance in air quality and meteorological data processing. Future work will explore more advanced physical mechanisms by incorporating more advanced physical principles to enrich the physical foundation of our model. We also plan to extend our model to a wider range of forecasting tasks and assess its generalizability.

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

# A  APPENDIX

## A.1  THE USE OF LARGE LANGUAGE MODELS

We used a general-purpose large language model (ChatGPT, OpenAI) only for language editing and LaTeX proofreading. The model suggested edits for grammar, wording, and clarity, and helped check equation formatting (e.g., LaTeX syntax, bracket matching, symbol consistency, and cross-references). It did not propose research ideas, design methods, write substantive technical content, generate or analyze data, or run experiments. All technical content and equations were authored by us and manually verified. The authors take full responsibility for all contents of the paper.

## A.2  TRANSFORMER MODEL

By leveraging self-attention mechanisms, Transformers can effectively capture long-range dependencies while allowing for parallel computation.

### A.2.1  ARCHITECTURE OF THE TRANSFORMER MODEL

Transformer models usually consist of two main components: Encoder and Decoder.

**Encoder**  The encoder consists of a stack of $L$ identical layers. Each layer contains two primary components: a multi-head self-attention mechanism and a position-wise feed-forward network. The input $X \in \mathbb{R}^{T \times d}$ is processed as follows:

$$\widetilde{X} = \text{LayerNorm}(X + \text{MultiHeadSelfAttention}(X)), \tag{11}$$

$$\text{EncoderOutput} = \text{LayerNorm}(\widetilde{X} + \text{FFN}(\widetilde{X})), \tag{12}$$

where $\text{FFN}(x) = \text{ReLU}(xW_1 + b_1)W_2 + b_2$ is applied to each position independently. Residual connections and layer normalization are used after each sub-layer to facilitate stable training and deep stacking.

**Decoder**   The decoder has a structure similar to the encoder but includes additional mechanisms to support sequence generation:

- Masked Multi-Head Self-Attention: Ensures that the decoder only attends to past positions in the sequence, preserving causality.
- Encoder-Decoder Multi-Head Attention: Allows the decoder to attend to the encoder's output, integrating source sequence information to generate target sequences.

The decoder employs a similar combination of residual connections and layer normalization.

### A.2.2   ATTENTION MECHANISM

The attention mechanism evaluates the relevance of each input element with respect to others in the sequence. Given Query ($Q$), Key ($K$), and Value ($V$) matrices, it assigns weights to positions based on their similarity.

**Scaled Dot-Product Attention**   The Scaled Dot-Product Attention computes weights by scaling the dot product of $Q$ and $K$ with $\sqrt{d_k}$ (the dimension of $K$) and applying the softmax function. The formula is:

$$\text{Attention}(Q, K, V) = \text{softmax}\left(\frac{QK^{\mathrm{T}}}{\sqrt{d_k}}\right)V, \tag{13}$$

where the scaling factor prevents large dot product values, ensuring stable gradients during training.

**Multi-Head Attention**   Multi-Head Attention extends Scaled Dot-Product Attention by using multiple attention heads to capture diverse patterns. It is computed as:

$$\text{MultiHead}(Q, K, V) = [\text{head}_1; \ldots; \text{head}_h]\,W^O, \tag{14}$$

where each head is defined as:

$$\text{head}_i = \text{Attention}(QW_i^Q, KW_i^K, VW_i^V), \tag{15}$$

and $W_i^Q, W_i^K, W_i^V$, and $W^O$ are learnable weight matrices. This design enables the model to focus on different subspaces of the input, enhancing its expressiveness.

### A.3   DATASET DETAILS

Table 3: Station information (Beijing)

| Station Name | Longitude | Latitude |
|---|---|---|
| Aotizhongxin | 116.397 | 39.982 |
| Changping | 116.23 | 40.217 |
| Dingling | 116.22 | 40.292 |
| Dongsi | 116.417 | 39.929 |
| Guanyuan | 116.339 | 39.929 |
| Gucheng | 116.184 | 39.914 |
| Huairou | 116.628 | 40.328 |
| Nongzhanguan | 116.461 | 39.937 |
| Shunyi | 116.655 | 40.127 |
| Tiantan | 116.407 | 39.886 |
| Wanliu | 116.287 | 39.987 |
| Wanshouxigong | 116.352 | 39.878 |

Table 4: Data description and units

| Data Name | Description | Unit |
|---|---|---|
| $PM_{2.5}$ | $PM_{2.5}$ concentration | $\mu g/m^3$ |
| $PM_{10}$ | $PM_{10}$ concentration | $\mu g/m^3$ |
| $SO_2$ | $SO_2$ concentration | $\mu g/m^3$ |
| $NO_2$ | $NO_2$ concentration | $\mu g/m^3$ |
| CO | CO concentration | $\mu g/m^3$ |
| $O_3$ | $O_3$ concentration | $\mu g/m^3$ |
| TEMP | temperature | $^\circ C$ |
| PRES | pressure | hPa |
| DEWP | dew point temperature | $^\circ C$ |
| RAIN | precipitation | mm |
| wd | wind direction | - |
| WSPM | wind speed | m/s |

Table 5: Station information (UK) (rows 1–30)

| Station Name | Longitude | Latitude |
|---|---|---|
| Inverness | -4.241 | 57.481 |
| Aberdeen Erroll Park | -2.095 | 57.157 |
| Edinburgh St Leonards | -3.182 | 55.946 |
| Greenock A8 Roadside | -4.734 | 55.944 |
| Glasgow Townhead | -4.244 | 55.866 |
| Glasgow High Street | -4.238 | 55.861 |
| Sunderland Silksworth | -1.407 | 54.884 |
| Hartlepool St Abbs Walk | -1.204 | 54.683 |
| Belfast Centre | -5.929 | 54.600 |
| Stockton-on-Tees A1305 Roadside | -1.316 | 54.566 |
| High Muffles | -0.809 | 54.334 |
| York Bootham | -1.087 | 53.968 |
| York Fishergate | -1.076 | 53.952 |
| Blackpool Marton | -3.007 | 53.805 |
| Leeds Centre | -1.546 | 53.804 |
| Preston | -2.680 | 53.766 |
| Hull Freetown | -0.341 | 53.749 |
| Dewsbury Ashworth Grange | -1.637 | 53.693 |
| Immingham Woodlands Avenue | -0.213 | 53.619 |
| Wigan Centre | -2.638 | 53.549 |
| Salford Eccles | -2.334 | 53.485 |
| Manchester Piccadilly | -2.238 | 53.482 |
| Glazebury | -2.472 | 53.460 |
| Sheffield Tinsley | -1.396 | 53.411 |
| Sheffield Barnsley Road | -1.456 | 53.405 |
| Sheffield Devonshire Green | -1.478 | 53.379 |
| Wirral Tranmere | -3.023 | 53.373 |
| Chesterfield Loundsley Green | -1.455 | 53.244 |
| Chesterfield Roadside | -1.457 | 53.232 |
| Crewe Coppenhall | -2.453 | 53.116 |

Table 6: Station information (UK) (rows 31–60)

| Station Name | Longitude | Latitude |
| --- | --- | --- |
| Wrexham | -3.003 | 53.042 |
| Stoke-on-Trent Centre | -2.175 | 53.028 |
| Nottingham Centre | -1.146 | 52.955 |
| Burton-on-Trent Horninglow | -1.636 | 52.821 |
| Telford Hollinswood | -2.437 | 52.673 |
| Leicester University | -1.127 | 52.620 |
| Norwich Lakenfields | 1.303 | 52.615 |
| Birmingham Ladywood | -1.918 | 52.481 |
| Birmingham A4540 Roadside | -1.875 | 52.476 |
| Coventry Allesley | -1.560 | 52.412 |
| Wicken Fen | 0.291 | 52.298 |
| Leamington Spa Rugby Road | -1.543 | 52.295 |
| Leamington Spa | -1.533 | 52.289 |
| Narberth | -4.692 | 51.783 |
| St Osyth | 1.049 | 51.778 |
| Oxford St Ebbes | -1.260 | 51.745 |
| Newport | -2.977 | 51.601 |
| Swindon Walcot | -1.766 | 51.558 |
| Southend-on-Sea | 0.678 | 51.544 |
| London Bloomsbury | -0.126 | 51.522 |
| London N. Kensington | -0.213 | 51.521 |
| London Hillingdon | -0.461 | 51.496 |
| London Harlington | -0.442 | 51.489 |
| Rochester Stoke | 0.635 | 51.456 |
| Reading New Town | -0.944 | 51.453 |
| Chilbolton Observatory | -1.438 | 51.150 |
| Charlton Mackrell | -2.683 | 51.056 |
| Southampton Centre | -1.396 | 50.908 |
| Portsmouth | -1.069 | 50.829 |
| Lullington Heath | 0.181 | 50.794 |

Table 7: Station information (UK) (rows 61–63)

| Station Name | Longitude | Latitude |
| --- | --- | --- |
| Honiton | -3.197 | 50.792 |
| Yarner Wood | -3.717 | 50.598 |
| Plymouth Centre | -4.142 | 50.372 |

Table 8: Station information (California) (rows 1–30)

| Station Name | Longitude | Latitude |
|---|---|---|
| $CA_40171\_122256$ | -122.256 | 40.171 |
| $CA_39762\_121840$ | -121.840 | 39.762 |
| $CA_39621\_119719$ | -119.719 | 39.621 |
| $CA_39541\_119747$ | -119.747 | 39.541 |
| $CA_39534\_122191$ | -122.191 | 39.534 |
| $CA_39522\_119795$ | -119.795 | 39.522 |
| $CA_39400\_119740$ | -119.740 | 39.400 |
| $CA_39189\_121999$ | -121.999 | 39.189 |
| $CA_39139\_121619$ | -121.619 | 39.139 |
| $CA_38746\_121265$ | -121.265 | 38.746 |
| $CA_38568\_121493$ | -121.493 | 38.568 |
| $CA_38202\_120680$ | -120.680 | 38.202 |
| $CA_37962\_121281$ | -121.281 | 37.962 |
| $CA_37682\_121441$ | -121.441 | 37.682 |
| $CA_37642\_120994$ | -120.994 | 37.642 |
| $CA_37488\_120836$ | -120.836 | 37.488 |
| $CA_37361\_118331$ | -118.331 | 37.361 |
| $CA_36953\_120034$ | -120.034 | 36.953 |
| $CA_36843\_121362$ | -121.362 | 36.843 |
| $CA_36819\_119716$ | -119.716 | 36.819 |
| $CA_36785\_119773$ | -119.773 | 36.785 |
| $CA_36314\_119644$ | -119.644 | 36.314 |
| $CA_36271\_115238$ | -115.238 | 36.271 |
| $CA_36214\_115091$ | -115.091 | 36.214 |
| $CA_36209\_121126$ | -121.126 | 36.209 |
| $CA_36173\_115333$ | -115.333 | 36.173 |
| $CA_36170\_115263$ | -115.263 | 36.170 |
| $CA_36142\_115079$ | -115.079 | 36.142 |
| $CA_36106\_115253$ | -115.253 | 36.106 |
| $CA_36049\_115053$ | -115.053 | 36.049 |

Table 9: Station information (California) (rows 31–60)

| Station Name | Longitude | Latitude |
| --- | --- | --- |
| CA$_3$6008$_-$115263 | -115.263 | 36.008 |
| CA$_3$5988$_-$115149 | -115.149 | 35.988 |
| CA$_3$5970$_-$114835 | -114.835 | 35.970 |
| CA$_3$5786$_-$115357 | -115.357 | 35.786 |
| CA$_3$5495$_-$120666 | -120.666 | 35.495 |
| CA$_3$5049$_-$118189 | -118.189 | 35.049 |
| CA$_3$4891$_-$120433 | -120.433 | 34.891 |
| CA$_3$4725$_-$118179 | -118.179 | 34.725 |
| CA$_3$4638$_-$120457 | -120.457 | 34.638 |
| CA$_3$4511$_-$117326 | -117.326 | 34.511 |
| CA$_3$4445$_-$119828 | -119.828 | 34.445 |
| CA$_3$4428$_-$119691 | -119.691 | 34.428 |
| CA$_3$4276$_-$118684 | -118.684 | 34.276 |
| CA$_3$4252$_-$119143 | -119.143 | 34.252 |
| CA$_3$4144$_-$117851 | -117.851 | 34.144 |
| CA$_3$4066$_-$118227 | -118.227 | 34.066 |
| CA$_3$3999$_-$117416 | -117.416 | 33.999 |
| CA$_3$3996$_-$117492 | -117.493 | 33.996 |
| CA$_3$3831$_-$117939 | -117.939 | 33.831 |
| CA$_3$3794$_-$118171 | -118.171 | 33.794 |
| CA$_3$3720$_-$116190 | -116.190 | 33.720 |
| CA$_3$3677$_-$117331 | -117.331 | 33.677 |
| CA$_3$3217$_-$117396 | -117.396 | 33.217 |
| CA$_3$2845$_-$117124 | -117.124 | 32.845 |
| CA$_3$2842$_-$116768 | -116.768 | 32.842 |
| CA$_3$2792$_-$115562 | -115.562 | 32.792 |
| CA$_3$2790$_-$116944 | -116.944 | 32.790 |
| CA$_3$2710$_-$117143 | -117.143 | 32.710 |
| CA$_3$2690$_-$114614 | -114.614 | 32.690 |
| CA$_3$2676$_-$115483 | -115.483 | 32.676 |

Table 10: Station information (California) (rows 61–62)

| Station Name | Longitude | Latitude |
| --- | --- | --- |
| CA$_3$2631$_-$117059 | -117.059 | 32.631 |
| CA$_3$2579$_-$116929 | -116.929 | 32.579 |