# OpenReview forum: "Diffusion-Advection Transformer for Air Quality Prediction"
_ICLR.cc/2026/Conference — ICLR 2026 Conference Desk Rejected Submission_

### Official Review · Reviewer_4pVz · 2025-10-22

**Soundness:** 2
**Presentation:** 2
**Contribution:** 3
**Rating:** 2
**Confidence:** 4

**Summary:**

The paper proposes DA-Transformer, a physics-informed spatiotemporal forecasting model for air quality (PM2.5) that embeds diffusion and advection operators directly inside Transformer encoder layers. In each encoder layer, Diffusion is implemented through a discrete Laplacian with a temperature-dependent coefficient, while advection module shifts the features along the direction and strength of local wind fields to simulate how pollutants are carried through the air.  The outputs of these two modules are then combined with learnable weights and added back into the Transformer’s residual connections. The model adopts learnable positional encodings, global spatiotemporal self-attention over flattened station–time tokens, and a simple non-autoregressive prediction head that maps the encoded window to multi-horizon outputs. Experiments on Beijing, UK (AURN), and a newly curated California dataset report consistent MAE/RMSE improvements over several baselines. Moreover, ablation studies suggest modest but positive contributions from the physics operators and positional encodings .

**Strengths:**

1. Integrating learnable diffusion and advection operators directly within Transformer encoder layers offers a distinct alternative to prior graph-based PDE solvers and neural ODE formulations. It avoids reliance on external solvers or predefined spatial graphs, and maintains the Transformer’s scalability and parallelism.

2. Introduces a coherent architecture combining the physics block, global attention, learnable positional encodings, and a non-autoregressive prediction head.

3. Experimental results across three geographical regions demonstrate consistent improvements over a diverse set of baselines, and ablation studies indicate that both the physics modules and positional encodings contribute to model performance.

**Weaknesses:**

1. **Decoder Inconsistency** :  The paper claims a non-autoregressive head that “removes the decoder stack,” (Section 3.1), yet Implementation Details state the model “consists of 3 encoder layers and 2 decoder layers.” This causes a discrepancy which should be resolved.

2. **Issues with Diffusion Formulation:**
- The diffusion operator (Eq. 5) applies the discrete Laplacian simultaneously along the station index and the feature-channel index.
While diffusion over the spatial axis is physically consistent with the PDE’s $D\nabla^{2}u$ term, diffusing along the feature dimension has no physical analogue in pollutant transport. Feature channels in $H^{(\ell)}$ represent heterogeneous or latent variables, not spatial coordinates. Therefore, smoothing across them mixes unrelated semantics. This turns the physics-motivated Laplacian into a numerical regularizer rather than a faithful discretization of spatial diffusion, undermining the claim of physical interpretability.

- The implementation assumes periodic boundaries along both station and feature axes, effectively connecting the last station back to the first in a cyclic topology. Such a condition is unrealistic for irregularly distributed monitoring networks, where geographic continuity does not prevail. In practice, pollutant dispersion at boundary stations should follow conditions like Dirichlet (fixed-value) or be modeled through a graph-based Laplacian reflecting true spatial adjacency. The periodic assumption can distort spatial relationships and introduce non-physical transport between distant or unconnected locations.

3. **Issues with the Advection Formulation:** In Section 3.4.2, Advection uses directional derivatives formed by  differences along station order and differences along the feature axis.

- The first derivative $(\delta_{\text{stat}} H)$, assumes that stations can be arranged on a 1-D chain based on nearest-neighbor distance. This simplification ignores the true 2-D (latitude–longitude) geometry of the monitoring network. It risks misaligning wind direction with station indices. Furthermore, Spatial directional dependence (e.g., north–south vs. east–west transport) cannot be captured in a single 1-D stencil.

- The second finite-difference term, $(\delta_{\text{feat}} H)_{b,s,i,c}$ treats the feature dimension $c$ as if it were a spatial coordinate, coupling it with wind components. However, $c$ indexes observed features (e.g. PM2.5, temperature or hidden embeddings), not spatial directions. Transporting information along feature channels with real wind velocities is physically meaningless: the wind field acts in geographic space, not across neural feature dimensions.

4. **Evaluation Weaknesses:**

- Baselines considered in the study are insufficient. Performance should be compared against Deep Learning based air quality prediction methods (PM25GNN), other transformer based (Airformer) and physics guided (Air-DualODE) models .
- It's better to provide hyperparameter details of baseline models for reproducibility in the Appendix.
- The data curation steps, splits, smoke episode handling, and code release details are insufficient in the California Dataset to be considered as a contribution.
- Ablations are too narrow. They do not test alternative discretizations, boundary conditions, or station-ordering choices. Given the physical concerns above, these ablations are critical to validating the approach. Moreover, the study omits attention ablations, even though the architecture heavily relies on both global spatial-temporal self-attention and the physics module.

5. Minor formatting weaknesses throughout the manuscript. (Ex: Some equations are not properly numbered) Please recheck and refine for clarity

**Questions:**

1. Is the final model encoder-only with a non-autoregressive head, or encoder–decoder? Please reconcile Sections 3.1 vs. Implementation Details, and state which variant produced Table 1/Table 2.

2. How robust is the station ordering by proximity under varying wind directions? Have you tried graph-based spatial derivatives (e.g., learned graph with edge directions aligned to wind)?

3. Have you conducted any efficiency analysis comparing DA-Transformer against standard encoder–decoder baselines?

---

### Official Review · Reviewer_azRE · 2025-10-25

**Soundness:** 2
**Presentation:** 2
**Contribution:** 1
**Rating:** 2
**Confidence:** 5

**Summary:**

This paper addresses the air pollution prediction task by proposing the Diffusion-Advection Transformer (DA-Transformer), a novel physics-informed architecture. The model integrates key physical mechanisms, specifically diffusion and advection, as differential equation-based components directly within a Transformer framework. This approach is designed to capture pollutant transport dynamics, such as local spreading and wind-driven transport, thereby grounding the model's predictions in physical principles. Notably, the methodology features a learnable discretization, modeling the diffusion term with a temperature-dependent coefficient and avoiding the use of external PDE solvers or predefined graphs. The effectiveness of the proposed model is evaluated through extensive experiments on three real-world datasets, where it is shown to outperform various baseline models.

**Strengths:**

- A strength of this work is its physically-informed approach to the diffusion component. The authors incorporate a temperature-conditioned diffusion block within each DA-Transformer layer.

- The paper's adoption of a learnable positional encoding strategy, as detailed in Section 3.2, is a commendable design choice. Instead of relying on fixed sinusoidal functions, the model assigns a trainable embedding vector to each time step, which is optimized jointly with the other parameters.

- The authors validate their model across three distinct real-world datasets (Beijing, UK, and California) , which represent a diverse range of emission and meteorological regimes.

**Weaknesses:**

- The overall novelty of the model architecture appears limited. The core idea of integrating diffusion and advection processes into a neural network framework follows a similar philosophy to prior physics-informed models, such as AirPhyNet. While the paper claims its distinction lies in embedding these operators directly into a Transformer without explicit graph construction, it does not provide a sufficient justification for the rationale of this specific architectural design.

- The paper's motivation for selecting a Transformer as the model backbone, instead of a Graph Neural Network (GNN), is not adequately substantiated. The paper critiques GNNs for the difficulty in defining an accurate graph structure , yet its own advection module relies on a simplified 1D ordering of stations by geographic proximity to approximate spatial gradients. This approach may fail to capture the complex 2D spatial dynamics. The paper does not convincingly demonstrate the necessity or superiority of using a Transformer's global attention mechanism over a GNN's structured message-passing for this physics-informed task.

- The learnable positional encoding, while presented as a contribution, raises significant concerns about its generalizability. The design, which assigns an independent, trainable embedding vector to each time step of the input sequence5, is inherently tied to a fixed input length ($S$). It is unclear how this model would handle input sequences of different lengths during inference. Furthermore, the paper does not specify how positional information is handled for the future time steps being predicted, which is a critical aspect of forecasting. This lack of flexibility for varying input lengths or generalizing to unseen future time steps is a notable limitation.

- The paper fails to provide any analysis of the model's computational complexity or practical time overhead.

**Questions:**

- What are the specific advantages of this paper compared to AirPhyNet? The overall design of the physics-guided network seems to follow the approach of AirPhyNet. The paper's justification for using a Transformer over a GNN framework like that in AirPhyNet is unclear . AirPhyNet's GCN-based architecture is also directly informed by advection-diffusion physics. The authors should elaborate on the specific advantages of their Transformer-based approach, especially given that they resort to approximating spatial gradients by ordering stations, whereas GNNs are naturally suited for such irregular spatial structures.

- The paper's claims regarding the flexibility of the learnable positional encoding  are not fully convincing. Since these embeddings are tied to specific time indices (1...S) and become fixed after training, it is unclear how they offer more flexibility for non-periodic events than a fixed encoding. Furthermore, this design lacks generalizability; the paper does not explain how the model would handle input sequences of varying lengths or how positional information is provided for the future prediction horizon.

- The ablation study  is too coarse-grained to validate the specific contributions of the physics modules. The study removes the advection and diffusion operators together rather than ablating them individually. Given that the temperature-dependent diffusion coefficient  is highlighted as a contribution, a more fine-grained ablation is necessary to specifically quantify the performance impact of this temperature-conditioning, separate from the advection component.

- The paper implements the physics modules as discrete operators and pairs this with a non-autoregressive prediction head. The rationale for abandoning continuous-time formulations (e.g., Neural ODEs), which are a natural fit for modeling differential equations, in favor of this specific discrete, one-shot approach is not discussed. The authors should clarify the advantages of this design choice over an ODE-based solver

- The related work section  appears brief. The review of physics-informed models, in particular, could be expanded to provide a more detailed comparison with existing methods, clarifying how the DA-Transformer's specific integration of physics differs from or improves upon other state-of-the-art approaches

- In the paper, it is noted that one of the issues with using GNNs in related work is the need for precise and fixed physical information. However, when employing differential approximation in this study, is such information still required? If not, how can the accuracy of this differential approximation be ensured?

- The paper critiques GNNs as being computationally expensive but fails to analyze the complexity of its own model. The self-attention mechanism results in a complexity that scales quadratically with the product of stations and time steps. The paper should provide a comparative analysis of computational efficiency, including training and inference times, to substantiate its claims.

---

### Official Review · Reviewer_RSdU · 2025-10-28

**Soundness:** 2
**Presentation:** 1
**Contribution:** 1
**Rating:** 2
**Confidence:** 4

**Summary:**

This paper proposes a novel physics-guided approach, DA-Transformer, for modeling air quality data. Specifically, it integrates the diffusion–advection phenomenon of pollutant transport into the Transformer architecture through a first-order discretization of the corresponding partial differential equations. Experiments are conducted on three representative datasets, including one newly constructed by the authors. The comparative results demonstrate that the proposed method outperforms the baselines.

**Strengths:**

1.A novel physics-guided approach based on the Transformer architecture is proposed, which differs from existing physics-guided methods in air quality prediction.

2.The Fig. 1 clearly illustrates the proposed method.

3.The DA-Transformer is evaluated on three distinct datasets, each representing a unique environmental setting.

**Weaknesses:**

1.The authors should clarify their motivation and the challenges of existing physics-guided approaches in the introduction. Currently, this paper gives the impression that the main goal is simply to incorporate physics-guided modules into a Transformer architecture, rather than to advance the modeling paradigm within the physics-guided domain.

2.The DA-Transformer introduces the discrete diffusion-advection equation in the latent space; however, the latent representation H is derived from pollutant and meteorological data. This raises a serious concern: does such a representation truly satisfy the physical assumptions of the diffusion-advection equation? This is a serious issue that undermines the physical validity of the model.

3.No evidence is given that such surrogate station-order-based derivatives approximate true spatial gradients. This could lead to physically incorrect modeling in irregular sensor layouts.

4.Although the authors define a discretized solution scheme to embed PDEs into the Transformer architecture, they only adopt a first-order discretization. While this choice is simple and computationally efficient, it raises concerns about potential cumulative errors compared to ODE-solver-based methods. It is similar to using Euler’s method to solve differential equations.

5.The related work section is insufficient, and the baselines listed in the main experiment table are quite limited. For example, AirFormer, a data-driven model specifically designed for air pollutant, is not mentioned or compared.

6.There is an inconsistency between the problem statement and the description in the methodology section. The output dimension of DA-transformer is confusing.

7.The novelty of this work is limited, as it primarily focuses on incorporating physics modules into the Transformer architecture without introducing fundamentally new modeling techniques.

8.The introduction does not effectively highlight the main contributions and is overly wordy.

**Questions:**

1.Could the authors explain why the ablation results vary across datasets when the diffusion–advection modules are removed? In particular, why are the performance differences on the UK and California datasets relatively small—are there other influencing factors specific to these regions?

2.MAPE is also an important metric for spatiotemporal forecasting; achieving improvements only in MAE and RMSE is not sufficiently convincing.

3.Could author provide more results on other spatiotemporal prediction baselines like Airformer[1], PM2.5GNN[2]?

4.Could the authors provide evaluation results for sudden change scenarios in Table 1? The definition of sudden changes can be referenced from AirFormer[1].

[1]. Yuxuan Liang, et al. AirFormer: Predicting Nationwide Air Quality in China with Transformers. AAAI 2023

[2]. Shuo Wang, et al. Pm2. 5-gnn: A domain knowledge enhanced graph neural network for pm2. 5 forecasting. ACM SIGSPATIAL 2020

---

### Official Review · Reviewer_32GY · 2025-10-30

**Soundness:** 2
**Presentation:** 1
**Contribution:** 2
**Rating:** 2
**Confidence:** 4

**Summary:**

This paper proposes DA-Transformer, a physics-informed Transformer model designed for spatiotemporal air quality forecasting. It explicitly integrates diffusion and advection mechanisms, key physical processes governing pollutant transport, directly into the Transformer encoder layers. Diffusion and Advection operators are differentiable and jointly learned with the model weights. The authors demonstrate the effectiveness of their approach across three diverse real-world datasets (Beijing, UK, California), showing consistent improvements over classical, deep sequence-based, Transformer-based, and graph-based baselines. The paper includes some ablation studies to assess the impact of the physics-informed modules and learnable positional encoding.

**Strengths:**

1. Instead of using external solvers or GNN-based PDE approximations, the paper proposes a clean integration of discrete diffusion/advection within Transformer layers.

2. Physics modules (e.g., temperature-based diffusion coefficients, wind-driven advection) are well grounded in the paper.

3. Multiple datasets with varying environmental regimes enhance credibility of generalizability claims.

**Weaknesses:**

1. The introduction section fails to summarize the core contributions of the paper and is unnecessarily verbose.

2. The authors should maintain consistent terminology throughout the paper—specifically, they should decide between using “physics-informed” (e.g., lines 20, 84) and “physics-guided” (e.g., line 455), as the two terms are not conceptually equivalent.

3. The paper includes insufficient baselines for spatiotemporal models.

4. The novelty of the paper is quite limited, as many of the proposed methods are adapted from existing work (e.g., learnable physics modules are commonly used in AirPhyNet[1] and Air-DualODE[2]), and therefore should not be claimed as original contributions.

**Questions:**

1. Could the authors provide a formal complexity analysis of the DA-Transformer as well as the other data-driven baselines (e.g., STGCN[3], AirFormer[4], AirPhyNet[1] and Air-DualODE[2])?

2. The authors compared several time-series models, but LSTM and Informer is already a relatively early work in Time Series Prediction. Could the authors include a comparison with more recent methods such as PatchTST[5]?

3. Could the authors provide a sensitivity analysis of some key hyperparameters?

4. Considering that sudden changes are crucial in air quality prediction, could the authors include corresponding comparison results in Table 1? This is a common practice in other air quality prediction studies.


[1]. AirPhyNet: Harnessing Physics-Guided Neural Networks for Air Quality Prediction.

[2]. Air Quality Prediction with Physics-Guided Dual Neural ODEs in Open Systems.

[3]. Spatio-temporal graph convolutional networks: a deep learning framework for traffic forecasting.

[4]. Airformer: Predicting nationwide air quality in china with transformers.

[5]. A time series is worth 64 words: Long-term forecasting with transformers.

---

### Note · Program_Chairs · 2026-01-17
**Submission Desk Rejected by Program Chairs**

The following references in this submission do not refer to real documents and/or have major errors in bibliographic information:

 Zheng Peng and Anuj Karpatne. Physics-guided deep learning for spatiotemporal forecasting: A survey. IEEE Transactions on Knowledge and Data Engineering, 2022. doi: 10.1109/TKDE. 2022.3192801.